# On the Role of Glycolysis in Early Tumorigenesis—Permissive and Executioner Effects

**DOI:** 10.3390/cells12081124

**Published:** 2023-04-10

**Authors:** Fabrizio Marcucci, Cristiano Rumio

**Affiliations:** Department of Pharmacological and Biomolecular Sciences, University of Milan, Via Trentacoste 2, 20134 Milan, Italy; cristiano.rumio@unimi.it

**Keywords:** glycolysis, tumor initiation, senescence, oncoprotein, DNA repair, chromatin

## Abstract

Reprogramming energy production from mitochondrial respiration to glycolysis is now considered a hallmark of cancer. When tumors grow beyond a certain size they give rise to changes in their microenvironment (e.g., hypoxia, mechanical stress) that are conducive to the upregulation of glycolysis. Over the years, however, it has become clear that glycolysis can also associate with the earliest steps of tumorigenesis. Thus, many of the oncoproteins most commonly involved in tumor initiation and progression upregulate glycolysis. Moreover, in recent years, considerable evidence has been reported suggesting that upregulated glycolysis itself, through its enzymes and/or metabolites, may play a causative role in tumorigenesis, either by acting itself as an oncogenic stimulus or by facilitating the appearance of oncogenic mutations. In fact, several changes induced by upregulated glycolysis have been shown to be involved in tumor initiation and early tumorigenesis: glycolysis-induced chromatin remodeling, inhibition of premature senescence and induction of proliferation, effects on DNA repair, *O*-linked *N*-acetylglucosamine modification of target proteins, antiapoptotic effects, induction of epithelial–mesenchymal transition or autophagy, and induction of angiogenesis. In this article we summarize the evidence that upregulated glycolysis is involved in tumor initiation and, in the following, we propose a mechanistic model aimed at explaining how upregulated glycolysis may play such a role.

## 1. Introduction

Oncogenesis is a multistep process whereby several genetic (mutations, amplifications, translocations), epigenetic or post-translational mechanisms eventually give rise to tumors with unlimited growth and, in most cases, metastatic potential. These oncogenic mechanisms lead to overexpression, constitutive activation or acquisition of novel biological activities (gain-of-function, GOF) of molecules (mostly proteins) endowed with oncogenic potential (oncoproteins) or to loss-of-function (LOF) of proteins with oncosuppressive potential (oncosuppressive proteins) [1,2]. In order to give rise to a growing and invasive tumor, the sequential accumulation over time of more than one change (e.g., overexpression of an oncoprotein or LOF of an oncosuppressive protein) [3,4] is required, because a single change is, in most cases, not sufficient to traverse all steps necessary to produce a full-blown tumor.

There is ample evidence, from preclinical as well as clinical investigations, that supports a multistep model of oncogenesis. Thus, for example, in sporadic colorectal cancer (CRC), LOF of the adenomatous polyposis coli (APC) tumor suppressor promotes the formation of a benign adenoma, but further oncogenic events are required for the progression to an invasive adenocarcinoma [5,6]. Increased epidermal growth factor receptor (EGFR) activity is causally linked to many types of cancer [7]. In *Drosophila* tumor models, the overexpression of EGFR alone leads to hyperplasia of the affected tissue, but without progressing to a malignant tumor. Only when combined with additional alterations do neoplastic transformation and metastatic dissemination occur [8].

Many tumor recurrence cases are associated with glycolysis, which is required for energy production (see Figure 1 for an overview of glycolytic metabolism, including some of the metabolic pathways that branch off and are mentioned in this article), even under oxygen-sufficient conditions (the so-called Warburg effect), instead of the more efficient mitochondrial respiration [tricarboxylic acid (TCA) cycle and oxidative phosphorylation (OXPHOS)] [9]. This metabolic reprogramming is now considered one of the hallmarks of cancer [1]. It should be stressed, however, that this phenomenon is neither complete nor universal as originally thought. Thus, a compilation study with data from 31 cancer cell lines or tissues showed that the average percentage of adenosine triphosphate (ATP) contribution from glycolysis was 17% [10]. These results imply that a relevant fraction of tumors rely mainly on mitochondrial respiration for energy generation [11]. Tumor cells that rely on glycolysis for energy production can coexist with tumor cells that rely on mitochondrial respiration, with the latter using glycolysis-derived lactate as fuel for energy generation [12,13]. Moreover, when glucose concentrations are low, tumor cells can switch from glycolysis to mitochondrial respiration [14]. Eventually, tumor cells may also exist in a hybrid metabolic state, with glycolysis and mitochondrial respiration coexisting in these cells, with recent studies showing that the TCA cycle is mainly used in the G1 phase and glycolysis in the S phase of the cell cycle [15,16]. It has even been argued that such a hybrid metabolic state may be particularly efficient in promoting tumor growth [17].

Given the relevance of glycolysis in tumorigenesis, two questions arise. First, why does a relevant fraction of tumors switch to glycolysis for energy production, either alone or together with mitochondrial respiration? Second, when does this occur during tumorigenesis? As to the first question, the acquisition of a glycolytic phenotype by certain tumors depends, at least in part, on the tissue of origin of the tumor [18], and lineage-specific transcription factors are involved in regulating tumor metabolism [19,20]. In addition, a glycolytic phenotype can be induced by stimuli from the tumor microenvironment (TME). Several of these stimuli, such as hypoxia or compressive stress, come into play when the tumor exceeds a certain size and the vascularization of the tumor becomes inadequate to deliver sufficient oxygen to all tumor cells [21,22]. In addition to hypoxia and mechanical forces, other stimuli generated in the TME or that diffuse into the TME have been shown to induce a glycolytic switch: bacterial infections [23], ultraviolet radiation [24], antitumor therapeutics [25], reactive oxygen species (ROS) [26], and high glucose concentrations [27] are examples of such stimuli. While these are stimuli that accompany tumor growth, there are now clear indications that tumor cell glycolysis is also involved in the earliest steps of tumorigenesis. In this article, we summarize the experimental evidence of this involvement and discuss mechanistic aspects. In particular, we address the effects of glycolytic metabolism on chromatin accessibility, i.e., the degree to which nuclear molecules (e.g., transcription factors) are able to interact with DNA; senescence, i.e., the process whereby cells stop dividing and undergo a permanent growth arrest without dying; DNA repair, i.e., the mechanisms whereby a cell identifies and rectifies damage caused to DNA molecules; *O*-linked β-*N*-acetylglucosamine (*O*-GlcNAc)ylation, the post-translational modification of serine and threonine residues of intracellular proteins with *O*-GlcNAc; apoptosis, i.e., a form of programmed cell death characteristic of multicellular organisms; epithelial–mesenchymal transition (EMT), i.e., the process whereby polarized epithelial cells acquire phenotypic traits typical of mesenchymal cells; autophagy, i.e., the process whereby a cell degrades self-components through a lysosome-dependent process; and angiogenesis, i.e., the process whereby new blood vessels are formed from preexisting vessels. 

Beforehand, however, we discuss some preliminary considerations. The acquisition of a glycolytic phenotype by tumor cells is accompanied by qualitative (e.g., post-translational modifications, expression of embryonic isoforms) and/or quantitative (overexpression of transporters or enzymes or overproduction of metabolites) changes in the glycolytic pathway [28]. When affecting glycolytic enzymes, these changes endow them with functions unrelated to their role in metabolism and are referred to as “moonlighting” functions [29]. Most of the effects of glycolysis that will be discussed in the following are, in fact, moonlighting functions. Second, the acquisition of a glycolytic phenotype by tumor cells does not necessarily imply that mitochondrial respiration is down-regulated and that lactate is the end-product. As we will see in the following, upregulated glycolysis can give rise to the production of lactate but also to an increased fueling of mitochondrial respiration [30]. Moreover, upregulated glycolysis can also increase the fueling of metabolites to anabolic metabolism, including the pentose phosphate pathway (PPP) (Figure 1), allowing cells to synthesize precursors for biological macromolecules (e.g., nucleotides, amino acids, and lipids) [31,32]. In the following, when we use the term glycolysis, we refer not only to “lactate-generating glycolysis” but also to the other outcomes.

## 2. Clinical Evidence for a Role of Glycolysis in Early Tumorigenesis

An important piece of evidence suggesting a role of glycolysis in early tumorigenesis comes from results obtained in epidemiological and clinical studies. First of all, clinical conditions characterized by hyperglycemia such as type II diabetes and obesity are associated with an increased incidence of several cancer types such as CRC, hepatocellular carcinoma, breast, renal, bladder, and pancreatic cancer [33,34]. Further evidence has been obtained in studies with clinical samples. Thus, studies with cervical cancer cell lines and human tissue samples of intraepithelial neoplasia and invasive squamous cell carcinoma showed that glucose transporter (GLUT) 1 was expressed from the earliest stages of cervical carcinogenesis and was a specific marker for dysplasia and carcinoma [35]. Increased expression of GLUT1 was paralleled by increased glucose uptake and an overall increase in glycolytic metabolism [36]. GLUT1 expression was also investigated in endometrial hyperplasia and endometrial carcinoma [37]. Proliferative endometrium, secretory endometrium, adenomyosis, and simple hyperplasia did not show GLUT1 expression. On the other hand, GLUT1 expression was observed in an increasing number of cases of complex hyperplasia without or with atypia and almost all cases of adenocarcinoma. GLUT1 expression increased at increasing distance of tumor cells from the stroma and the blood vessels. This suggests that hypoxia was at the origin of increased GLUT1 expression. Similar results were obtained with endometrial tissues, with GLUT1 expression increasing from endometrial hyperplasia to carcinoma [38]. A progressive increase in GLUT1 expression from normal tissues, over hyperplastic tissue, low-grade and high-grade dysplasia, up to carcinomas, was also observed in gallbladder tissues; normal tissues scored negative. Of note, strong GLUT1 expression was highly specific for detecting gallbladder carcinomas [39]. Wellberg et al. [40], on the other hand, showed that GLUT1 expression was higher in ductal carcinoma in situ compared to normal breast tissue but lower in invasive versus in situ lesions, suggesting that the requirement for GLUT1 was predominant in early cancer lesions but decreased with tumor progression [40].

Barrett’s esophagus is a premalignant condition with chronic acid bile reflux. This causes tissue damage and is accompanied by periodic episodes of ischemia and hypoxia. Using four cell lines derived from patients with Barrett’s esophagus, it was shown that the cell lines from patients with more advanced and genetically unstable Barrett’s esophagus had up to two-fold higher glycolytic metabolism compared to a cell line from a patient with earlier, genetically stable Barrett’s esophagus [41]. All cell lines had active mitochondria and upregulated OXPHOS upon inhibition of glycolysis. These results suggest that the periodic episodes of hypoxia played a causative role in upregulating glycolysis without compromising mitochondrial respiration.

Keloids are benign fibroproliferative cutaneous lesions that may predispose patients to several tumors, in particular skin cancer [42]. Keloid fibroblasts undergo metabolic reprogramming from OXPHOS to glycolysis with the upregulation of glycolytic enzymes [43]. This is accompanied by higher glucose influx and lactate production compared to normal fibroblasts. Importantly, the proliferation of keloid fibroblasts is suppressed upon the inhibition of glycolysis with 2-deoxy-glucose (2DG).

Overall, these results suggest that glycolysis becomes upregulated in preneoplastic and neoplastic lesions with the expression, in most but not all cases, constantly increasing until full-blown cancer. In some cases, glycolysis upregulation appeared to be the consequence of episodes of hypoxia affecting cells undergoing neoplastic transformation.

A mechanistic insight into how preneoplastic lesions can undergo metabolic reprogramming has recently been reported [44]. In this study, cells with low glycolytic metabolism were used, similar to those present in early lesions of ductal carcinoma in situ (DCIS) of the breast. In this cancer, tumor cells grow toward the ductal lumen at increasing distance from the blood vessels. This leads to oxygen deprivation, causing the cells to rely on glycolysis for energy production. Consequently, lactate is produced, and an acidic TME is generated. Damaghi et al. [44] exposed breast cancer cells with low glycolytic metabolism for several months to a combination of hypoxia, acidic pH, low glucose, and starvation. At the end of the study, single clones were isolated for characterization. The harshest conditions led to the selection of clones relying on aerobic glycolysis (Warburg effect). The transcription factor Krüppel-like factor 4 (KLF4) was identified as a potential inducer of the Warburg effect. Moreover, in DCIS samples, KLF4 expression was strongest in areas that were likely affected by the harshest environmental conditions. KLF4 had been previously identified as an essential factor for pluripotent stem cell development and the induction of aerobic glycolysis [45,46]. Overall, these results confirm that tumor cells can upregulate their glycolytic metabolism at early stages of malignant progression and that microenvironmental stimuli, including hypoxia, may play an important role in the acquisition of this phenotype.

## 3. Oncoproteins and Other Oncogenic Molecules Are Potent Upregulators of Glycolysis

A clear indication that glycolysis is not only upregulated in response to stressors from the TME comes from the finding that oncoproteins are potent upregulators of glycolysis. Oncoproteins can cooperate with stimuli from the TME in shaping the glycolytic phenotype of tumor cells [47,48]. This finding also suggests that a glycolytic switch accompanies or is involved in the neoplastic transformation step that is induced by a given oncoprotein, including the earliest steps of tumorigenesis. In this section, we discuss the oncoproteins and other oncogenic molecules that induce the upregulation of glycolysis. Several reviews have been published over the years on this topic, e.g., [49]. For this reason, we will only briefly address the main classes of oncoproteins and other oncogenic molecules and discuss in some more detail how these molecules lead to glycolysis upregulation and the role that it plays in early tumorigenesis.

### 3.1. Oncoproteins or Oncometabolites

The number of oncoproteins that have been shown to upregulate glycolysis is very large and includes, among others, oncogenic viruses [50,51,52], oncogenic KRAS [30,53,54,55] and BRAF [56], epidermal growth factor receptor (EGFR) [57,58] and human epidermal growth factor receptor (HER) 2/erythroblastic oncogene 2 (ErbB2) [31], the phosphoinositide 3-kinase (PI3K)-AKT-mechanistic target of rapamycin (mTOR) pathway [59,60,61,62], the DEK oncoprotein [63], the Wnt/β-catenin pathway [64], transcription factors such as hypoxia-inducible factor (HIF)-1 [65], c-Myc [36,66], signal transducer and activator of transcription (STAT) 3 [67], and epithelial–mesenchymal transition (EMT) master transcription factors like zinc finger E-box-binding homeobox (ZEB-1) [68].

In most of these cases, glycolysis upregulation is the consequence of the overexpression of one or more glycolytic enzymes. There are, however, exceptions to this rule. This is when glycolytic enzymes undergo a posttranslational modification or an interaction with other molecules that endows them with enhanced activity. Thus, it has been shown that one of two isoforms of the oncogene KRAS, KRAS4A, undergoes a direct, guanosine triphosphate (GTP)-dependent interaction with hexokinase (HK) 1 that increases its enzymatic activity [69]. This interaction takes place upon colocalization of KRAS4A with HK1 on the outer mitochondrial membrane. Another example is represented by the oncogene proviral insertion in murine lymphomas 2 (PIM2), a serine/threonine kinase, that upregulates tumor cell glycolysis by directly phosphorylating pyruvate kinase (PK) M2 on Thr454 [70].

Since many of these oncoproteins represent tumorigenic stimuli, the observation that they upregulate glycolysis suggests that this metabolic pathway may be involved in the neoplastic transformation process itself, independently of its effects on growing tumors. In fact, there are examples that support this assumption. Thus, in a model of EGFR-driven epithelial neoplasia in *Drosophila*, lactate dehydrogenase (LDH) was upregulated and required for the transition from hyperplasia to neoplasia. Further, in a human primary breast cell culture model, LDH promoted the acquisition of a transformed phenotype [58]. LDH isoform A (LDHA) overexpression was required for c-Myc-mediated transformation in c-Myc-transformed Rat1a fibroblasts, c-Myc-transformed human lymphoblastoid cells and Burkitt lymphoma cells [66,71]. An increased LDHA level was required for the growth of a transformed spheroid cell mass with a hypoxic internal microenvironment. A reduction of LDHA levels abrogated the soft agar clonogenicity of Myc-transformed cells but not anchorage-dependent tumor cell growth, suggesting that elevated LDHA expression was required for the acquisition of a neoplastic phenotype. Using a switchable model of Myc-driven liver cancer, it was found that tumor LDHA levels and activity as well as several glycolysis pathway genes increased and preceded tumor formation. These changes also paralleled the in vivo conversion of pyruvate to lactate. On the other hand, they were rapidly inhibited as tumors started to regress, suggesting that these metabolic changes were caused by the activity of a single oncoprotein (Myc) [72]. The ubiquitously expressed protein 14-3-3ζ plays an important role in the regulation of several signaling pathways involved in tumor initiation and progression and is overexpressed in many tumor types. Its overexpression in human mammary epithelial cells (hMEC) has been shown to upregulate glycolysis and LDHA expression and to promote early transformation [73]. Knockdown of LDHA in the 14-3-3ζ-overexpressing hMECs reduced glycolytic activity and inhibited transformation. LDHA upregulation was the result of the 14-3-3ζ-mediated activation of the mitogen-activated protein kinase kinase (MEK)-extracellular signal-regulated kinase (ERK)-cAMP-response element binding protein (CREB) signaling pathway [73].

In a mouse model of glioblastoma multiforme, EGFR activation induced the translocation of PKM2 into the nucleus [74]. Here, PKM2 bound to phosphorylated β-catenin. The complex then bound to the *Ccnd1* promoter, and this led to histone deacetylase (HDAC) 3 removal from the promoter, histone H3 acetylation, and cyclin D1 expression. The PKM2-β-catenin interaction was a necessary event for EGFR-driven tumor cell proliferation and brain tumor development [74].

As regards the PI3K-AKT-mTOR pathway, one study reported that mTOR upregulated glycolysis due to the HIF-1α-mediated transcription activation of glycolytic enzymes and Myc–heterogeneous nuclear ribonucleoprotein-dependent regulation of PKM2 gene splicing [60]. Importantly, this article showed that the disruption of PKM2 suppressed oncogenic mTOR-mediated tumorigenesis, suggesting that PKM2 upregulation was a necessary event in mTOR-driven tumorigenesis.

In mouse models of KRAS-driven lung cancer and HER2-driven breast cancer, HK2 was necessary for tumor initiation and maintenance as demonstrated using *Hk2* conditional knockout mice [31]. HK2 deletion was accompanied by a reduced diversion of glucose through the non-oxidative branch of the PPP and reduced synthesis of ribonucleotides, reduced diversion into the serine biosynthesis pathway, and less efficient fueling into the TCA cycle. The upregulation of glycolysis in mice expressing activated HER2 was found to be required for the development of mammary tumors in these mice [40].

The deletion of the long non-coding RNA nuclear paraspeckle assembly transcript (NEAT) 1 in mouse mammary tumor virus-polyoma middle tumor-antigen (MMTV-PyVT) mice has been shown to impair tumor initiation, growth, and metastasis [75]. NEAT1 acted as a scaffold for the assembly of three glycolytic enzymes, i.e., phosphoglycerate kinase (PGK) 1/phosphoglycerate mutase (PGM) 1/enolase (ENO) 1. The formation of this complex promoted substrate channeling, thereby boosting glycolytic metabolism. *Neat1*-proficient mice developed large tumors within 130 days. These were poorly differentiated and aggressive adenocarcinomas. On the other hand, *Neat1*-deficient mice developed small tumors with hyperplasia-like, non-malignant features that were poorly vascularized. These results suggest that NEAT1 exerts its pro-tumorigenic role through the upregulation and optimization of glycolytic metabolism [75].

Hereditary leiomyomatosis and renal cell cancer is due to the biallelic inactivation of the gene encoding the Krebs cycle enzyme fumarate hydratase. This leads to the accumulation of the metabolite fumarate, which inactivates the factors that are involved in the replication of mitochondrial DNA, thereby leading to the loss of respiratory chain components, and, consequently, promoting a shift to aerobic glycolysis. This shift is responsible for disease progression in this aggressive cancer. The shift to aerobic glycolysis is also accompanied by diversion of glucose to the PPP [76]. Thus, in this case, the impairment of mitochondrial respiration leads to the upregulation of glycolytic metabolism, which plays a tumorigenic role. 

### 3.2. Oncosuppressive Proteins

Molecules endowed with oncosuppressive properties can become tumorigenic if, due to mutations, they lose their oncosuppressive function (LOF) or can even become directly oncogenic if they acquire new functions (GOF). The prototypic example of this class of proteins is the transcription factor p53. P53 has been shown to downregulate the expression of GLUTs [77,78] and inhibit glycolysis through various mechanisms, such as the inhibition or inactivation of enzymes directly or indirectly involved in the regulation of glycolytic metabolism [79,80,81]. In addition, p53 is a positive regulator of mitochondrial metabolism and is involved in cellular redox control [82]. Importantly, p53 can acquire GOF mutations, thereby not only losing its capacity to inhibit glycolysis, but even acquiring the capacity to promote it [83,84]. The genetic inactivation of *Tp53* in neural progenitor/stem cells has been shown to give rise to high-grade gliomas and this was accompanied by a glycolytic switch [85]. Importantly, in another model, the inhibition of glycolysis impaired mutant p53-driven tumorigenesis [86]. In this case, mutant p53 stimulated glycolytic metabolism, promoting GLUT1 translocation to the plasma membrane. Evidence that p53 loss and upregulated glycolysis may represent important steps in early tumorigenesis has also been suggested by Kawauchi et al. [78]. They showed that in p53-deficient primary cultured cells, the activity of nuclear factor kappa-light-chain-enhancer of activated B cells (NF-κB) was enhanced and this caused an increase in glycolysis and the upregulation of GLUT3. Ras-induced cell transformation and the upregulation of glycolysis in these p53-deficient cells were suppressed in the absence of p65/NF-κB expression and were restored by GLUT3 expression. These results point to an essential role of upregulated glycolysis in this model of Ras-induced cell transformation.

Another protein with oncosuppressive effects that are lost upon LOF mutations is phosphatase and tensin homolog (PTEN), a negative regulator of the PI3K-AKT-mTOR pathway. PTEN with a LOF mutation led to the upregulation of GLUTs and glycolytic enzymes [87]. Interestingly, two different oncosuppressors with LOF mutations have been shown to cooperate in promoting tumor cell glycolysis. Thus, the combined loss of PTEN and p53 in prostate cancer upregulated HK2 through an increase in mRNA translation (induced by *Pten* deletion) and an increase in mRNA stability (mediated by *Tp53* deletion) [88]. Upregulated HK2, in turn, induced aerobic glycolysis, which was necessary for the growth of *Pten*/*Tp53*-deficient tumors in xenograft mouse models of prostate cancer.

Breast cancer type 1 susceptibility protein (BRCA1) is another oncosuppressive protein, which, upon LOF mutations, is associated with an increased risk of developing different types of cancer. Its silencing in ovarian surface epithelial and fallopian tube cells increased HK2 expression and glycolysis, an effect mediated by Myc and STAT3 [89]. 

Together, the examples that have been listed in the present and the previous sections show that the oncoprotein- or oncosuppressive protein-driven upregulation of glycolytic metabolism plays, in the models that have been discussed, a necessary role for early tumorigenesis, thereby supporting the clinical observations that have been described in Section 2.

## 4. Can Upregulated Glycolysis Play a Causative Role in Oncogenesis?

In the previous section, we discussed that oncoproteins can upregulate glycolysis and that this can play a necessary role during the initial steps of tumorigenesis. At this point, one is led to ask whether upregulated glycolysis acts only as a mediator of tumor growth, even if at early stages of tumorigenesis, or if upregulated glycolysis can act by itself as an instigating factor for oncogenesis. There is evidence that the latter can, indeed, be the case, and we discuss it in this section. Evidence discussed in greater details in this section is synoptically summarized in Table 1.

Morris et al. [90], while investigating the energetic requirements of intestinal stem cells (ISC) of *Drosophila* flies, found that these cells met the enhanced energetic requirement in response to proliferative stimuli through the increased uptake of mitochondrial Ca^++^. Increased Ca^++^ uptake, in turn, enhanced mitochondrial respiration, which generated the energy needed to sustain ISC proliferation. In old flies, however, the uptake in mitochondrial Ca^++^ declined, thereby leading to metabolic reprogramming towards aerobic glycolysis (Warburg effect). This reprogramming induced a hyperplasia of ISCs that was very similar to that induced by the oncogene Ras^V12^ [89]. These results suggest that upregulated glycolysis may act as a tumorigenesis-inducing signal similar to that of Ras^V12^ which requires at least one other signal (e.g., mutation or overexpression of an oncoprotein) in order to induce a full-blown tumor. Similar conclusions were drawn earlier based on a different model [91]. Here, the oncoprotein RCL, a N-glycoside hydrolase, and LDHA were shown to induce, synergistically, the anchorage-independent growth of Rat1a fibroblasts [91]. Cells expressing both RCL and LDHA formed tumors when injected subcutaneously into nude mice, but cells expressing one or the other alone did not induce tumorigenesis. Both *Rcl* and *Ldha* are c-Myc target genes, and the same authors previously showed that LDHA overexpression is required for c-Myc-induced transformation because lowering of the LDHA expression reduced the soft agar clonogenicity of c-Myc-transformed fibroblasts and lymphoma cell lines [66]. In another work, the knockdown of LDHA stimulated mitochondrial respiration and greatly diminished the tumorigenicity of HER2-driven mammary tumor cells [92]. Some cell lines switched to an OXPHOS-based metabolism, while other cell lines were strictly LDHA-dependent. Most recently, it has been reported that LDHA can have an oncogenic effect in breast cancer cells by binding to the active, small GTPase Rac1, thereby inhibiting its interaction with its negative regulator and keeping Rac1 in its active state [93]. In clinical breast cancer tissues, LDHA overexpression was associated with higher Rac1 activity.

The evidence discussed so far shows that upregulated glycolysis can play a causative or co-causative role in tumorigenesis by acting through different mechanisms of action. In addition, it would also be interesting to know whether glycolysis itself can be at the origin of tumor-driving mutations and/or upregulate the expression or activity of oncoproteins. This first possibility was investigated by Yun et al. [93]. These authors showed that GLUT1 was upregulated in CRC cell lines with mutated *KRAS* or *BRAF* genes. These cell lines had upregulated glucose uptake and glycolysis and survived under low glucose conditions, conditions in which only cells with mutant *KRAS* or *BRAF* genes survived. These results are consistent with those discussed before in that they suggest that oncoproteins upregulate glycolysis, and this conferred a growth and survival advantage to the cells. In addition, the authors showed that the selection of cells with wild-type *KRAS* alleles for proliferation in low glucose medium led to a heritable upregulation of GLUT1 in the majority of surviving clones. Importantly, a sizeable fraction of the surviving clones (4.4%) had mutations in *KRAS* [94]. This finding suggests that upregulated glycolysis may have played a causative role in the acquisition of these mutations. It should be stressed, however, that only a minority of clones had acquired *KRAS* mutations, suggesting that other undefined factors were required for the induction of *KRAS* mutations in a larger fraction of clones. Further evidence suggesting that glycolysis may contribute to the acquisition of tumor-driving mutations and contribute to tumor initiation has been provided more recently [95]. These authors showed that the earliest stages of CRC initiation were characterized by glycolytic metabolism and showed downregulation of the mitochondrial pyruvate carrier (MPC). This is a protein heterodimer in the inner mitochondrial membrane that transports pyruvate into the mitochondrial matrix for oxidative metabolism. The authors also showed in two different CRC mouse models that loss of *Mpc1* led to upregulated glycolysis, increased expression of ISC markers, and increased frequency of adenoma formation with the generation of higher-grade tumors in response to a tumorigenic stimulus. Moreover, the appearance of hyperproliferative lesions in response to the homozygous loss of *Apc* was accompanied by low expression of MPC. On the other hand, overexpression of MPC with increased pyruvate import blocked tumor formation. Further results suggested that *Mpc1* loss and upregulated glycolysis led to an increased frequency of *Apc* loss of heterozygosity, the crucial genetic tumorigenic alteration in this model, thereby promoting further mutational events facilitating the shift from a benign adenoma into an invasive adenocarcinoma. These are important results suggesting that *Mpc1* loss and, consequently, upregulated glycolysis played a causative role in the generation of CRC driver oncogenes and tumor initiation.

As regards the glycolysis-induced upregulation of the expression or activity of oncoproteins, GLUT3 and glycolytic metabolism have been shown to induce the activation of NF-κB, thereby facilitating RAS-induced cell transformation [78]. PKM2 was found to phosphorylate the mTORC1 inhibitor AKT1 substrate 1 (AKT1S1) [96]. This promoted the activation of mTORC1 signaling independent of exogenous stimuli and led to accelerated tumor growth. In fact, the stimulus-independent, constitutive activation of mTORC1 promotes a switch from catabolic to anabolic metabolism, a hallmark of unrestrained tumor cell proliferation and tumor growth in vivo [97]. Further, unchecked STAT3 activity facilitates tumor formation, and the overexpression of STAT3 is sufficient to transform 3T3 fibroblasts and induce tumor formation in different mouse models [98,99]. PKM2 has been shown to increase STAT3 activity and this facilitated the generation of anaplastic large cell lymphoma [100]. Nuclear PKM2 enhanced STAT3 activity in CRC cells, leading to the increased resistance of these cells to gefitinib [101]. It should also be noted that STAT3 induced PKM2 phosphorylation in liver precancerous lesions, suggesting the possibility that a positive feedback between STAT3 and PKM2 may occur in malignant transformation [67].

A mechanism similar to that leading to increased STAT3 activity has been reported for PFK1 and the transcription factors Yes-associated protein (YAP)/transcriptional coactivator with PDZ-binding motif (TAZ). Under physiological conditions, YAP/TAZ regulate organ growth, but once their activity becomes unrestrained, they promote tumorigenesis and the maintenance of cancer stem cells (CSC) in different tumor types [102,103,104]. Enzo et al. [105] found that in cells that showed enhanced uptake of glucose, PFK1 associated with TEA domain family members (TEAD), which are cofactors of the YAP/TAZ transcription factors. This association stabilized the interaction of TEADs with YAP/TAZ in the nucleus, thereby enhancing the transcriptional activity of YAP/TAZ and, consequently, the progression of breast cancer. Moreover, human mammary tumors showed that glycolysis-regulated gene expression paralleled YAP/TAZ activity and correlated with worse prognosis and with the CSC content of the tumors [105]. 

The glycolytic metabolite lactate has also been shown to modulate the expression of oncoproteins and oncosuppressors. Thus, exogenous lactate upregulated the transcription of 673 genes in L6 cells [106]. More recently, it was found that lactate enhanced the transcription of key oncogenes (*MYC*, *RAS*, and *PI3KCA*), transcription factors (*HIF1A* and *E2F1*), tumor suppressors (*BRCA1*, *BRCA2*) as well as cell cycle and proliferation genes involved in breast cancer (*AKT1*, *ATM*, *CCND1*, *CDK4*, *CDKN1A*, *CDKN2B*) [107]. Both endogenous, glucose-derived lactate as well as exogenous lactate supplementation were able to do so. 

So far, we have only briefly mentioned CSCs. The term CSC probably includes true tumor-initiating cells, and this is certainly pertinent to the present topic but also established tumor cells that have undergone an EMT and acquired a phenotype allowing them to repopulate a tumor [108]. This latter case is not within the scope of the present article. Nevertheless, CSCs, whether belonging to the first or second class, often, but not always (see below for some exceptions), rely on glycolytic metabolism for their maintenance [109]. In mouse models of non-small cell lung cancer (NSCLC) driven by oncogenic KRAS or EGFR, LDHA was essential for CSC survival and proliferation [57]. GLUT1 was essential for the maintenance of pancreatic, ovarian, and glioblastoma CSCs, and a specific GLUT1 inhibitor inhibited the functionality of CSCs and, in vivo, it delayed tumor initiation after the implantation of CSCs [110]. Similar observations were reported for MCF-10A cells, a commonly used human mammary epithelial cell line for the study of normal breast cell function and transformation. In these cells, activated mTOR promoted metabolic reprogramming towards glycolysis [111]. This led to enhanced lactate production and an increased self-renewal potential and clonogenic power of the cells. This latter result is of particular interest because it suggests that it is the glycolytic switch per se to induce the acquisition of a CSC phenotype and a marker of neoplastic transformation. Very recently, it has been shown that 6-phosphofructo-2-kinase/fructose-2,6-biphosphatase 3 (PFKFB3) plays a crucial role in the maintenance of malignant pleural mesothelioma CSCs, and the PFKFB3 inhibitor PFK158 reduced CSC-mediated xenograft growth in vivo [112].

Overexpressed HK2 has been found to promote, upon its nuclear localization, a stem cell phenotype in both acute myeloid leukemia (AML) and normal hematopoietic cells [113]. This was the consequence of HK2 becoming phosphorylated and having interacted with nuclear proteins known to regulate chromatin accessibility, leading to increased chromatin accessibility at DNA domains promoting stemness and DNA repair. This led to decreases of double-strand breaks (DSB) and acquisition of chemoresistance. PKM2 was found to interact with octamer-binding transcription factor (OCT)-4 and enhanced OCT-4-mediated transcription [114]. OCT-4 is a transcription factor involved in the maintenance of the pluripotent potential of embryonic stem cells and CSCs, but not normal human tissues. An effect of PKM2 on stem cell transcription factors has also been shown by Lin Y et al. [115]. Thus, the knockdown of PKM2 combined with ionizing radiation reduced the expression of several CSC transcription factors, such as NANOG, OCT-4, sex determining region Y-box 2, and B lymphoma Mo-MLV insertion region 1 homolog.

**Table 1 cells-12-01124-t001:** Evidence for a causative role of glycolysis in tumorigenesis.

Biological Effects Observed	Role in Tumorigenesis	Ref.
ISCs of aging *Drosophila* flies met energetic demands by upregulating glycolysis. This led to ISC hyperplasia, similar to that induced by oncogenic RAS^V12^.	Oncogenic effect	[90]
RCL oncoprotein and LDHA induced the anchorage-independent growth of Rat1a fibroblasts. Cells expressing both RCL and LDHA formed tumors in mice, but one or the other alone was not tumorigenic.	Co-causative role in tumorigenesis	[91]
LDHA can keep the small GTPase Rac1 in an active state, thereby promoting the growth of breast cancer cells in vitro and in vivo.	Indirect oncogenic effect	[93]
Culturing CRC cells with wild-type *KRAS* alleles in low glucose medium caused the heritable upregulation of GLUT1 in the majority of surviving clones, and 4.4% had *KRAS* mutations.	Co-causative role in the acquisition of oncogenic mutations	[94]
CRC initiation was associated with the downregulation of MPC; this caused the upregulation of glycolysis and increased frequency of *APC* loss of heterozygosity. This facilitated the acquisition of further mutations and the progression of adenoma into invasive adenocarcinoma.	Co-causative role in the acquisition of oncogenic mutations.	[95]
GLUT3 and glycolysis induced the activation of NF-κB and this facilitated the RAS-induced cell transformation.	Activation of an oncoprotein.	[78]
PKM2 phosphorylated the mTORC1 inhibitor AKT1S1 and this facilitated the activation of mTORC1 signaling, leading to accelerated tumor growth.	Activation of an oncoprotein	[96]
PKM2 increased STAT3 activity, thereby facilitating the generation of ALCL	Activation of an oncoprotein	[100]
Nuclear PKM2 increased STAT3 activity and this increased the resistance of CRC cells to gefitinib.	Activation of an oncoprotein	[101]
PFK1 associated with TEADs and stabilized their interaction with YAP/TAZ, promoting the transcriptional activity of YAP/TAZ and breast cancer progression.	Activation of an oncoprotein	[105]
In KRAS- and EGFR-dependent mouse models of NSCLC, LDHA was essential for CSC survival and proliferation.	Promotion of the survival and expansion of CSCs.	[57]
GLUT1 was essential for the maintenance of CSCs from different tumor types. A GLUT1 inhibitor inhibited self-renewal, tumor-initiating capacity of CSCs, and delayed tumor initiation after in vivo implantation of CSCs.	Promotion of the survival and tumor-initiating capacity of CSCs	[110]
In human mammary epithelial MCF-10A cells, activated mTOR upregulated glycolysis. The increased lactate that was produced increased the self-renewal and clonogenic power of the cells.	Induction of the generation and expansion of CSCs	[111]
PFKFB3 was involved in the maintenance of malignant pleural mesothelioma CSCs. PFKFB3 inhibition diminished CSC-mediated xenografts in vivo.	Promotion of CSC survival	[112]
Upregulated HK2 localized to the nucleus of AML stem cells, where it was phosphorylated and promoted chromatin remodeling, increasing accessibilities at DNA domains, promoting stemness and DNA repair.	Induction of CSCs	[113]
PKM2 interacted with OCT-4 and increased OCT-4-mediated transcription.	Induction and maintenance of CSCs	[114]
Knockdown of PKM2 and ionizing radiation reduced the expression of several CSC transcription factors and markers.	Induction and maintenance of CSCs	[115]

Abbreviations: AKT1S1, AKT1 substrate 1; ALCL, anaplastic large cell lymphoma; AML, acute myeloid leukemia; APC, adenomatous polyposis coli; CRC, colorectal carcinoma; CSC, cancer stem cell; EGFR, epidermal growth factor receptor; HK, hexokinase; GLUT, glucose transporter; ISC, intestinal stem cell; LDHA, lactate dehydrogenase A; MPC mitochondrial pyruvate carrier; mTOR, mechanistic target of rapamycin; NSCLC, non-small cell lung cancer; OCT, octamer-binding transcription factor; PFK1, phosphofructokinase 1; PFKFB3, 6-phosphofructo-2-kinase/fructose-2,6-biphosphatase 3; PKM2, pyruvate kinase M2; STAT3, signal transducer and activator of transcription; TAZ, transcriptional coactivator with PDZ-binding motif; TEAD, TEA domain family members; YAP, Yes-associated protein.

## 5. Mechanistic Aspects of the Involvement of Glycolysis in Early Tumorigenesis

In the previous sections, we have discussed available evidence showing that oncoproteins can upregulate glycolysis from the earliest stages of tumorigenesis and that glycolysis itself may instigate tumorigenesis. At this point, the obvious question arises as to how, mechanistically, upregulated glycolysis contributes to early tumorigenesis. We will address this question here. Table 2 gives a concise summary of the evidence that will be discussed in this section. Figure 2 depicts the mechanisms involved in tumorigenesis induction, the enzymes or metabolites that act as mediators, and the final effects that are induced in cells undergoing the tumorigenic process.

### 5.1. Induction of Epigenetic Changes

As already discussed in the beginning, epigenetic modifications are now considered a fundamental mechanism leading to the expression of oncoproteins and/or inhibition of oncosuppressive proteins. Glycolytic metabolism was found to increase chromatin accessibility by inducing global histone acetylation, thereby facilitating DNA repair and increasing resistance to DNA-damaging drugs [116]. Along similar lines, it has been shown that pituitary tumor cells overexpressing GLUT1 undergo a metabolic reprogramming towards glycolysis when cultured at high glucose concentrations. The upregulation of glycolytic metabolism then led to an increase in histone acetylation (in particular at H3K9). This effect, in turn, increased the expression of telomerase reverse transcriptase (TERT) and, consequently, the proliferation of pituitary tumor cells [117]. Lactate may also play an important role, because it has been shown to derivatize histone lysine residues (“lactylation”) and this, in turn, increased chromatin accessibility [118]. It has been suggested that the histone lactylation-induced increase in chromatin accessibility may promote the expression of oncoproteins driving neoplastic transformation. Histone *O*-GlcNacylation has also been shown to influence the chromatin structure and regulate gene expression [119]. Eventually, the effect of nuclear HK2 leading to an increase in open chromatin conformation has been discussed before [113].

### 5.2. Inhibition of Premature Senescence

Indefinite proliferation is a hallmark of cancer [1]. A single oncoprotein (e.g., KRAS), however, is, in most cases, not sufficient to induce indefinite proliferation because the cell activates mechanisms to prevent it [120]. Cellular senescence, in addition to its accompanying cell growth arrest, is now recognized to play such an antiproliferative effect. KRAS, for example, leads, in the pancreas, to the formation of premalignant lesions that do not further advance to a malignant, invasive cancer because the cells start to undergo senescence. Thus, a single oncogenic signal is, in most cases, not sufficient to endow cells with unlimited growth potential. A second signal is required to bypass the senescence-induced blockade of proliferation, and upregulated glycolysis has been shown to provide such a signal.

Thus, in a study aimed at identifying genes that immortalize fibroblasts, PGM was identified as one such gene [79]. The upregulation of PGM activity enhanced glycolytic flux and promoted indefinite proliferation. Glucose 6-phosphate isomerase (GPI) was found to have similar effects. The knockdown of the expression of either enzyme triggered premature senescence. Later on, the same authors showed that the unlimited proliferative potential of embryonic stem cells correlated with upregulated glycolysis [121]. On the other hand, differentiating embryonic stem cells showed a decrease in glycolysis, suggesting that upregulated glycolysis was directly related to their replicative potential. Another study showed that the overexpression of the glycolytic enzyme glyceraldehyde 3-phosphate dehydrogenase (GAPDH) allowed the bypassing of the senescence response to the oncoprotein BRAF^V600E^ [56]. HK2 has also been identified as a senescence-bypassing molecule [122], as under conditions of HK2 upregulation, a pathway downstream of the HK2 metabolite glucose 6-phosphate, the hexosamine biosynthetic pathway (HBP), was shown to be involved in bypassing oncogene-induced senescence. Thus, the inhibition of glutamine fructose-6-phosphate amidotransferase, the first enzyme of the HBP, induced premature senescence, while GlcNAc allowed cells to bypass it. Since the HBP branches into N-linked glycosylation and *O*-GlcNAcylation, it is tempting to speculate that this mechanism of senescence-bypassing joins the one that will be described in Section 5.4. Lactate was also shown to bypass oncogene-induced senescence [123]. It did so by inducing the EMT-master transcription factor Snail, which inhibited p16^INK4a^ expression, thereby promoting senescence avoidance. These results suggest the intriguing possibility of the involvement of EMT in glycolysis-induced early tumorigenesis. Interestingly, another EMT master transcription factor, Twist, has been reported to collaborate with *Ras* in bypassing senescence [124]. It would be interesting to know whether glycolytic enzymes or metabolites are also involved in this model of oncogene-induced senescence.

Overall, there is considerable evidence that upregulated glycolysis may rescue cells from oncoprotein-induced senescence and, by so doing, allow cells to undergo indefinite proliferation. 

### 5.3. Effects on DNA Repair and the Cell Cycle

DNA lesions give rise to gene mutations and chromosomal damage, which, as already discussed in the beginning, are among the most frequent events at the origin of neoplastic transformation [125]. In the presence of limited levels of DNA damage, mechanisms leading to DNA and cell survival are activated, while high levels of DNA damage can induce cell death, either autophagy or senescence induced. Moreover, unrestrained proliferation burdens the cells with high metabolic demands. When these demands are met, at least in part, by mitochondrial respiration, then ROS accumulate and, consequently, further DNA damage occurs.

Upregulated glycolysis has been shown to have important, yet not univocal effects on damaged DNA, either promoting DNA repair (in most cases) or DNA damage (in a minority of cases). PFKFB3 has been suggested to play a role in promoting DNA damage repair (DDR), as an inhibitor of PFKFB3, PFK158, suppressed endometrial cancer cell proliferation, enhanced the sensitivity of the cells to carboplatin and cisplatin-induced DNA damage, and favored cell death, either apoptosis- or autophagy-mediated [126]. Moreover, PFKFB3 inhibition reduced glucose uptake, ATP production, and lactate release. Mechanistically, others showed that PFKFB3 localized into nuclear foci after the induction of DSBs by irradiation [127]. Here, PFKFB3 supported DNA synthesis during DNA repair by promoting the generation of a local pool of deoxynucleotide triphosphates (dNTP) at the site of DNA damage. PGM1 was also shown to act through a similar mechanism of action. Its securing of the intracellular dNTP pool facilitated the homologous repair of DSBs in cancer cells caused by DNA-damaging agents [128]. The DDR-promoting effect of PKM2, on the other hand, was a consequence of its phosphorylation by ataxia teleangectasia mutated (ATM) and subsequent nuclear localization [129]. Phosphorylated PKM2 led to the phosphorylation of the CtBP-interacting protein (CtIP). This led to increased recruitment of CtIP at DSBs and the consequent promotion of HR-mediated DDR. The inhibition of this pathway increased the susceptibility of cancer cells to a DNA-damaging agent and poly (ADP-ribose) polymerase (PARP) 1 inhibition [129]. 

PGK1 has also been suggested to have DDR-promoting effects, as its knockdown reduced the expression of proteins involved in DDR and methylation and lowered the cellular methylation level [130]. DDR-promoting effects have also been reported for lactate. Cancer cell lines selected for growth under glucose-deprived conditions were exposed to lactate and their response to cisplatin-induced DNA damage was investigated [131]. In cells exposed to lactate cisplatin showed reduced efficacy, less DNA damage, and increased expression of DNA repair genes.

DDR-promoting effects of both upregulated glycolysis and glutaminolysis have been demonstrated in irradiated MCF7 breast cancer cells [132]. Both pathways were shown to play a role in promoting DSB repair and preventing senescence after irradiation. This was the consequence of histone methylation upstream of the polycomb repressive complexes (PRC) 1/2, which are of crucial importance for non-homologous end-joining repair (NHEJR). Histone methylation was promoted by the HBP metabolite N-acetyl-glucosamine and *O*-GlcNAcylation [132].

So far, we have discussed findings suggesting that upregulated glycolysis promotes DDR. There are, however, also some results suggesting that glycolytic enzymes may promote DNA damage. For example, nuclear PKM2 has been reported to increase DNA damage and chromosomal aberrations in tumor cells exposed to etoposide [133]. Upregulated aldolase (ALDO) B induced the functional loss of mismatch repair (MMR) proteins in colon adenocarcinoma cell lines [134], thereby leading to irreversible DNA damage and inducing apoptosis. In fact, colon adenocarcinoma patients with high ALDOB mRNA expression had longer overall survival. 

### 5.4. Induction of O-GlcNAcylation of Target Proteins

We have already discussed an article showing that *O*-GlcNAcylation can promote histone methylation and facilitate a DDR [132]. Onodera et al. [135] demonstrated the additional involvement of *O*-GlcNAcylation promoted by upregulated glycolysis in early tumorigenesis. They used a 3-dimensional cell culture model that led to the overexpression of GLUT3 in nonmalignant breast cells. Such overexpression activated oncogenic signaling pathways, including EGFR, β1 integrin, MEK, and AKT, while the reduction of glucose uptake reversed this effect. The induction of the tumorigenic effects was mediated by PKM2 interacting with soluble adenylyl cyclase and subsequent activation of Ras-related protein 1 (RAP1), as well as the *O*-GlcNAcylation of several proteins. The inhibition of any one of the enzymes of the HBP leading to the generation of the substrate (UDP-GlcNAc) required for the *O*-GlcNAcylation of proteins arrested cell proliferation [135]. By the same token, hyperglycemia, a well-known risk factor for cancer, has been shown to induce the aberrant *O*-GlcNAcylation of many intracellular proteins that are involved in tumorigenic mechanisms and other pathological conditions, including cell proliferation, resistance to apoptosis, increased cell migration and invasiveness, epigenetic regulation, resistance to chemotherapy, and altered metabolism [136,137,138].

### 5.5. Induction of Antiapoptotic Effects

Apoptosis resistance is now considered one of the hallmarks of cancer [1]. It protects cells from potentially lethal insults from the TME. Upregulated glycolytic enzymes have been shown to exert antiapoptotic effects. Thus, HK2 has been reported to suppress both apoptosis and oxidative metabolism [139]. Mechanistically, the aberrant activation of the PI3K-AKT-mTOR pathway promoted translocation of HK2 to the outer mitochondrial membrane where it interacted with the permeability transition pore, which includes the voltage-dependent anion channel (VDAC) and B-cell lymphoma 2 (Bcl2)-associated X protein (Bax). This interaction led to the inhibition of apoptosis and promoted cell survival [140,141,142]. PKM2, under conditions of oxidative stress, was also shown to be directed to the mitochondria where it phosphorylated and stabilized Bcl2, thereby promoting apoptosis inhibition [143]. An antiapoptotic effect has also been reported for PFKFB3, a positive regulator of glycolysis. Its enzymatic product, fructose 2,6-biphosphate (F2,6BP), activated cyclin-dependent kinases (CDK) and induced the CDK-mediated phosphorylation of the Cip/Kip protein p27 [144]. This promoted the ubiquitination and proteasomal degradation of p27, which is a potent suppressor of the G1/S transition and an activator of apoptosis [144]. 

**Table 2 cells-12-01124-t002:** Mechanisms underlying the effects of glycolysis in tumor initiation and early tumorigenesis.

Mechanism	Effects Observed	Ref.
Induction of epigenetic changes	Induction of histone acetylation → stimulation of an open chromatin structure	[116]
	Glycolytic pituitary tumor cells → induction of histone acetylation → increased expression of TERT → stimulation of tumor cell proliferation	[117]
	Induction of histone lactylation → stimulation of an open chromatin structure → induction of the expression of oncoproteins	[118]
	Nuclear HK2 in leukemic SCs → stimulation of an open chromatin structure → promotion of DDR and chemoresistance	[113]
Inhibition of premature senescence.	Upregulation of PGM or GPI activity in fibroblasts →upregulation of glycolysis → induction of indefinite proliferation. Knockdown of PGM or GPI activity in fibroblasts → induction of premature senescence.	[79]
	Upregulated glycolysis and downregulated mitochondrial respiration in embryonic SCs → correlation with unlimited replicative potential. Downregulated glycolysis → differentiation of embryonic SCs.	[121]
	Upregulation of GAPDH → bypassing senescence induced by the melanoma oncoprotein BRAF^V600E^.	[56]
	Upregulation of HK2 → increased production of F6P →fueling of HBP → bypassing oncogene-induced senescence and growth arrest.	[122]
	Lactate → induction of EMT master transcription factor Snail → targeting and inhibition of the expression of p16^INK4a^→ bypassing of senescence.	[123]
Promotion of DNA repair	Inhibition of PFKFB3 in endometrial cancer cells → inhibition of glycolysis, suppression of proliferation, increased sensitivity and DNA damage to chemotherapeutics, increased cell death.	[126]
	Irradiation and induction of DSBs → PFKFB3 relocalized into nuclear foci → generation of a local pool of dNTP → support of DNA synthesis during DSB repair.	[127]
	PGM1 → securing of the intracellular pool of dNTP →DDR in response to DNA-damaging agents in cancer cells.	[128]
	Phosphorylation of PKM2 by ATM → nuclear accumulation of PKM2 → phosphorylation of CtIP → increased CtIP’s recruitment at DSBs and resection of DNA ends → promotion of HR-mediated DNA DSB repair.	[129]
	Knockdown of PGK1 → reduced expression of DDR-related proteins, methylation-related enzymes, and total cellular methylation level.	[130]
	Lactate on cancer cells selected to grow in glucose-deprived conditions → reduced efficacy of cisplatin, reduced signatures of DNA damage, enhanced DNA recombination competence and increased expression of DNA repair genes involved in restoring DNA damage.	[131]
	Upregulation of glycolysis and glutaminolysis in irradiated MCF7 breast cancer cells → increased production of HBP metabolite GlcNAc and *O*-GlcNAcylation → histone methylation upstream of PRCs 1/2 which are important for NHEJR.	[132]
Promotion of DNA damage	Nuclear PKM2 → increased DNA damage and chromosomal aberrations in tumor cells exposed to etoposide.	[133]
	ALDOB upregulated in CRC cell lines → induction of functional loss of MMR proteins → irreversible DNA damage and induction of apoptosis.	[134]
O-GlcNAcylation of target proteins.	3D cell culture model leading to overexpression of GLUT3 in nonmalignant breast cells: → activation of oncogenic signaling pathways → interaction between PKM2 and sAC → loss of cell polarity and increased cell proliferation; → upregulation of HBP → O-GlcNAcylation and regulation of the expression of β1 integrin, EGFR, and GLUT3 → loss of cell polarity and increased cell proliferation.	[135]
	Hyperglycemia → Induction of aberrant *O*-GlcNAcylation of proteins involved in tumorigenesis, including cell proliferation, resistance to apoptosis, increased cell migration and invasiveness, epigenetic regulation, resistance to chemotherapy and altered metabolism.	[137,138]
	Upregulation of glycolysis and glutaminolysis in irradiated MCF7 breast cancer cells → increased production of the HBP metabolite GlcNAc and O-GlcNAcylation → histone methylation upstream of the PRCs 1/2 which are important for NHEJR.	[132]
Induction of antiapoptotic effects	HK2 → suppression of apoptosis and oxidative metabolism.	[139]
	Aberrant activation of PI3K-AKT-mTOR → translocation of HK2 to the outer mitochondrial membrane → interaction with the permeability transition pore, including the VDAC and Bax → inhibition of apoptosis and promotion of cell survival.	[140,141,142]
	Oxidative stress → redirection of PKM2 to mitochondria → phosphorylation and stabilization of Bcl-2 → inhibition of apoptosis.	[143]
	F2,6BP → activation of CDKs → CDK-mediated phosphorylation of the Cip/Kip protein p27 → ubiquitination and proteasomal degradation of p27, a suppressor of the G1/S transition and activator of apoptosis.	[144]

Abbreviations: ALDOB, aldolase isoform B; ATM, ataxia teleangectasia mutated; Bax, Bcl-2-associated X protein; CDK, cyclin-dependent kinase; CRC, colorectal cancer; CtIP, CtBP-interacting protein; DDR, DNA damage repair; DSB, double-strand break; dNTP, deoxynucleotide triphosphates; EGFR, epidermal growth factor receptor; EMT, epithelial–mesenchymal transition; F2,6BP, fructose 2,6-biphosphate; F6P, fructose 6-phosphate; GAPDH, glyceraldehyde 3-phosphate dehydrogenase; GLUT, glucose transporter; GPI, glucose 6-phosphate isomerase; HBP, hexosamine biosynthetic pathway; HK, hexokinase; HR, homologous repair; MMR, mismatch repair; GlcNAc, N-Acetylglucosamine; NHEJR, non-homologous end-joining repair; *O*-GlcNAcylation, O-linked GlcNAc modification; PFKFB3, 6-phosphofructo-2-kinase/fructose-2,6-biphosphatase 3; PGM, phosphoglycerate mutase; PGK, phosphoglycerate kinase; PKM2, pyruvate kinase M2; PRC, polycomb repressive complex; sAC, soluble adenylyl cyclase; SC, stem cell; TERT, telomerase reverse transcriptase; VDAC, voltage-dependent anion channel.

### 5.6. Induction of EMT and Autophagy

There are several other effects induced by glycolytic enzymes and metabolites (e.g., induction of EMT/autophagy and angiogenesis) which, by and large, appear to be related to tumor growth rather than early tumorigenesis. Nevertheless, some evidence reported in the literature suggests that these effects may also be involved in early tumorigenesis. In the following, we will briefly discuss this evidence.

Upregulated glycolysis can induce EMT or macroautophagy (hereafter autophagy) in tumor cells. EMT as well as autophagy can be induced by several glycolytic enzymes as well as lactate and the resulting acidic TME. We have extensively reviewed this topic and discussed the signals that dictate whether glycolysis induces EMT or autophagy in tumor cells and, in some cases, even both [145,146,147,148]. EMT represents a phenotypic switch that occurs in tumor cells that have already undergone neoplastic transformation [108]. Autophagy, on the other hand, has been suggested to play a tumor-inhibitory effect during the initial stages of tumorigenesis, while having a tumor-promoting effect in more advanced stages because it allows tumor cells to survive in response to tumor cell-intrinsic or -extrinsic stressors [148]. There is some evidence, however, suggesting that EMT and autophagy may also play a role in early tumorigenesis.

As regards EMT, in a previous section, we discussed results showing that EMT master transcription factors may be involved in bypassing oncoprotein-induced senescence and that glycolytic components may induce one or more of these EMT master transcription factors that then promote senescence escape [123]. In addition, the EMT master transcription factor ZEB1 has been shown to upregulate the expression of PFKM by binding to a non-classic ZEB1-binding sequence in the promoter region of PFKM [68]. The silencing of ZEB1 in vitro and in vivo and investigations of the Cancer Genome Atlas (TCGA) database demonstrated the crucial role of the ZEB1-PFKM axis in carcinogenesis and metastasis. 

As regards autophagy, it is not known at which time point during tumor growth autophagy stops being tumor suppressive and starts becoming tumor promoting or if the two effects may overlap and coexist at some time point. For this reason, glycolysis-induced autophagy may well play a promoting role even during early tumorigenesis when tumor cells start becoming exposed to microenvironmental stressors, and autophagy may allow survival under these conditions [149]. 

### 5.7. Induction of Angiogenesis

Angiogenesis is one of the hallmarks of cancer and is an absolute requirement for tumors to grow beyond a certain size [150]. While angiogenesis cannot be considered an early tumorigenic event, it is necessary for a tumor to become clinically relevant. Lactate has been shown to have angiogenesis-promoting effects [151]. It does so indirectly, by activating tumor-associated macrophages (TAM) to adopt a pro-angiogenic phenotype. Such TAMs cooperate with tumor cells in the formation of new blood vessels and are considered a cause of resistance against anti-angiogenic therapies. In a recent article, it was shown that the overexpression of PKM2 in nodular urothelial hyperplasia with angiogenesis strongly accelerated tumorigenesis [152]. Mechanistically, this was the result of PKM2 binding to STAT3, the translocation of the complex into the cell nucleus, and the activation of the expression of angiogenic factors, including vascular endothelial growth factor and HIF-1α. Interestingly, the overexpression of PKM2 in simple urothelial hyperplasia did not trigger tumorigenesis, suggesting that PKM2 was involved at an amplification stage of tumorigenesis, involving angiogenesis.

## 6. A Mechanistic Model for Glycolysis-Induced Tumorigenesis

In the previous sections, we have listed several mechanisms of action that underlie the role of glycolysis in early tumorigenesis. In this section, we propose a model that aims to accommodate several of these different mechanisms within a unifying framework.

Initially, we discussed how oncoproteins, whether mutated or overexpressed, can upregulate glycolysis in tumor cells and how such upregulation may represent a necessary step for tumorigenesis. In the following, we discussed results showing that upregulated glycolysis can, by itself, represent a tumorigenic signal which, in combination with other signals, can lead to the development of full-blown tumors. Moreover, upregulated glycolysis has also been shown to promote the acquisition of mutations, thereby facilitating the generation of oncogenes and the expression of oncoproteins. While oncoproteins can upregulate glycolysis, one is led to ask about the stimuli that upregulate glycolysis in preneoplastic cells in the absence of any (apparent) involvement of oncoproteins. One possibility that has been proposed is that persistently upregulated glycolysis may represent an adaptation to intermittent hypoxia [153]. The knowledge that Barret’s esophagus, a premalignant condition characterized by periodic episodes of hypoxia, displays upregulated GLUT1 expression and glycolysis [41] is consistent with this view. In accordance with this hypothesis, it is possible that other intermittent stimuli arising in the microenvironment and known to upregulate glycolysis [21,22,23,24,25,26,27,145] may play a similar role. 

Following upregulation, we propose that a key mechanism whereby glycolysis promotes tumorigenesis lies in the epigenetic modifications that it can induce with some of its metabolites such as pyruvate or lactate (see Figure 3 for a summary view of the model herein proposed). Such changes include histone acetylation and lactylation, which then induce changes in chromatin conformation [116,118]. These changes may not represent all-or-nothing events, as it has been shown that increasing rates of glycolysis induce increasing rates of histone acetylation [154]. Consequently, different levels of changes of chromatin conformation may be expected at different levels of epigenetic modifications. Moreover, epigenetic modifications are not global changes that affect the molecular targets (e.g., histones) in a random manner. Thus, it has been shown that p300 histone acetylation in response to changing acetyl-CoA concentrations occurs in a specific, non-random fashion, with acetylation at some sites increasing while at others decreasing [155]. Some recent findings obtained with CRC samples are fully consistent with these results [156]. Thus, somatic mutations affecting the epigenome were found to lead to differential (loss or gain) chromosome accessibility of transcription factors. Interestingly, increased chromatin accessibility was observed for some transcription factors involved in development, such as the HOX, FOX, and SOX families.

We suggest that metabolite-induced epigenetic modifications and changes in chromatin conformation are permissive effects for several other tumor-promoting mechanisms that have been described before. We refer to these permissive effects as signal 1 of the tumorigenic actions of glycolysis. Thus, one of the effects of upregulated glycolysis is the bypassing of tumor cell senescence. A key event in the induction of senescence is preventing E2F transcription factors from gaining access to DNA due to the recruitment of heterochromatin-forming proteins and the generation of transcriptionally silent chromatin domains [157]. We propose that upregulated tumor cell glycolysis may induce, through its metabolites, chromatin remodeling so to prevent the formation or disrupt existing transcriptionally inactive chromatin domains, thereby allowing DNA accessibility to senescence-bypassing transcription factors such as E2F. DDRs, which are facilitated by upregulated glycolysis as discussed in Section 5.3, rely on an open chromatin conformation and it has been shown, for example, that the transcription factor Sal-like protein 4, which maintains self-renewal of embryonic stem cells, can induce drug resistance by enhancing DDRs through the upregulation of GLUT1 and glycolysis and, consequently, the induction of an open chromatin structure [158]. Given these results, it is tempting to speculate that, in this case, glycolysis also has a permissive effect (signal 1) allowing chromatin remodeling so to enable the recruitment of components of the DDR machinery leading to DNA repair. Glycolysis-induced chromatin remodeling may also have a permissive effect in the acquisition of potentially oncogenic mutations. Thus, while mutations relying on base substitutions appear to be more frequent in regions of closed chromatin, insertion- and deletion (indels)-based mutations and substitutions appear to occur more frequently in regions of open chromatin [159]. This suggests the possibility that glycolytic metabolite-induced chromatin remodeling may facilitate the acquisition of mutations leading to the expression of oncoproteins [93,118].

Overall, we suggest that upregulated glycolysis leads, through its metabolites, to epigenetic modifications that promote specific chromatin remodeling that is dependent on the level of epigenetic modification effected, thereby creating, eventually, permissive conditions for the induction of biological effects mediated by glycolytic enzymes or metabolites. These effects were discussed in Section 5 (e.g., bypassing of senescence, induction of a DDR and acquisition of mutations, anti-apoptotic effects). We refer to these effects as signal 2. Thus, it is the combination of the two signals that will eventually give rise to tumorigenic effects.

On the basis of the considerations outlined in this and the previous section, we propose the model depicted in Figure 4. First, glycolysis can be upregulated, on one hand, by intermittent hypoxia and/or other stress stimuli generated in the microenvironment. Glycolytic enzymes and/or metabolites can then induce permissive (signal 1) and/or executioner (signal 2) signals in order to instigate tumorigenesis. These effects include the generation of mutated proteins that can act as oncoproteins. Oncoproteins, independently of how they are generated, can then upregulate glycolysis and this will give rise to an amplifying feed-forward loop. 

We also suggest that upregulated glycolysis may not be the only inducer of permissive and executioner effects (signal 1 and signal 2, respectively). In fact, in at least in one case, glycolysis was formally excluded as being involved in tumor (leukemia) initiation [160]. On the other hand, several other mediators that can induce chromatin remodeling and promote permissive effects have been described in the literature, e.g., [161,162,163,164,165]. Moreover, executioner effects such as those described before (i.e., inhibition of senescence, promotion of a DDR, acquisition of mutations or overexpression of proteins, etc.) can be brought about by stimuli other than glycolytic enzymes [166,167]. For this reason, we believe that the 2-signal model for early tumorigenesis that we have proposed may be of more general validity, beyond that involving upregulated glycolysis through its enzymes and metabolites. This 2-signal and reciprocal amplification model is also compatible with the knowledge that oncogenesis is a multistep process that may take decades before developing into a full-blown tumor [168]. Moreover, some recent publications have reported results that are fully consistent with the proposed model, i.e., the convergence of permissive effects acting on chromatin accessibility and executioner effects (e.g., oncogenic mutations, bypassing senescence, etc.) in driving oncogenesis [156,169].

## 7. Conclusions

In this article, we have summarized available evidence on the role of glycolysis in early tumorigenesis. There is now considerable evidence that glycolysis, through its enzymes and metabolites, can instigate tumorigenesis from its earliest stages through different mechanisms of action. We have also aimed to outline a 2-signal model, based on permissive and executioner effects, that aims at accommodating, in a mechanistic framework, several of the tumor-promoting effects of glycolysis. We have also proposed that this model may be of more general validity, being applicable to other mechanisms underlying neoplastic transformation and tumorigenesis. 

While the results discussed so far demonstrate that upregulated glycolytic metabolism can play a causative role in early tumorigenesis in an undefined fraction of tumors, the question arises whether other metabolic pathways can also play a similar role. While a detailed discussion of this aspect is beyond the scope of the present article, there are clear indications that this, indeed, can be the case. Thus, overexpression of the enzyme glycine decarboxylase and other enzymes of glycine/serine metabolism have been found to promote transformation and tumorigenesis in normal primary fibroblasts and epithelial cells [170]. Moreover, leukemic stem cells derived from CML patients relied on upregulated oxidative metabolism for their survival [171]. Similar observations have been made with stem cells of other tumor types (reviewed in [172]). Mitochondrial respiration was also found to be essential for tumorigenesis in a KRAS-driven mouse model of lung cancer through the generation of ROS, which were required for the MAPK signaling-dependent induction of anchorage-independent growth [173]. We have already mentioned that the accumulation of fumarate may promote a shift to aerobic glycolysis [76]. Moreover, fumarate may also play a direct oncogenic role by inhibiting PTEN through a post-translational modification and, consequently, leading to the uncontrolled activation of the PI3K-AKT-mTOR pathway [173]. Further, other metabolites, generated through diverse metabolic pathways, including the herein discussed lactate, have been shown to play an oncogenic role through diverse mechanisms of action, including alterations of chromatin accessibility [163,174,175]. These few indications demonstrate that overexpressed enzymes and overproduced or otherwise accumulated metabolites from different metabolic pathways other than glycolysis can play a tumor-promoting role in early tumorigenesis. At this point, the question arises as to when and why a given metabolic pathway plays an instigating role in early tumorigenesis. Is it dependent on the tissue of origin of the tumor as we have discussed in the beginning and/or on the tumor type or does it depend on different classes of microenvironmental stimuli that promote the overexpression of enzymes of one metabolic pathway rather than the other? Answering these questions is of crucial importance both for the deepening of our understanding of fundamental processes of oncogenesis as well as for possible therapeutic applications that may derive from this knowledge.

## Figures and Tables

**Figure 1 cells-12-01124-f001:**
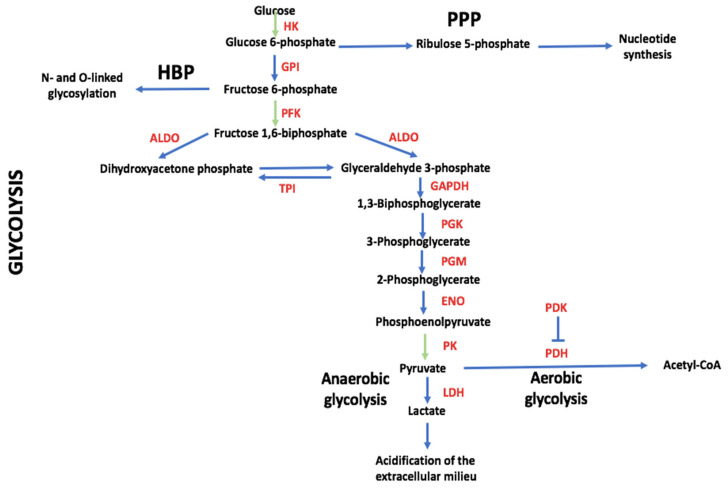
Glycolytic Metabolism. Enzymes and metabolites are depicted as well as some of the pathways that branch off from glycolysis and are discussed in the text. Green arrows indicate irreversible reactions. The final steps occurring under aerobic (aerobic glycolysis) or anaerobic conditions (anaerobic glycolysis) are indicated. ALDO, fructose-biphosphate aldolase; ENO, enolase; GAPDH, glyceraldehyde 3-phosphate dehydrogenase; GPI, glucose 6-phosphate isomerase; HBP, hexosamine biosynthetic pathway; HK, hexokinase; LDH, lactate dehydrogenase; PDH, pyruvate dehydrogenase; PDK, pyruvate dehydrogenase kinase; PFK, phosphofructo-2-kinase; PGK, phosphoglycerate kinase; PGM, phosphoglycerate mutase; PK, pyruvate kinase; PPP, pentose phosphate pathway; TPI, triosephosphate isomerase.

**Figure 2 cells-12-01124-f002:**
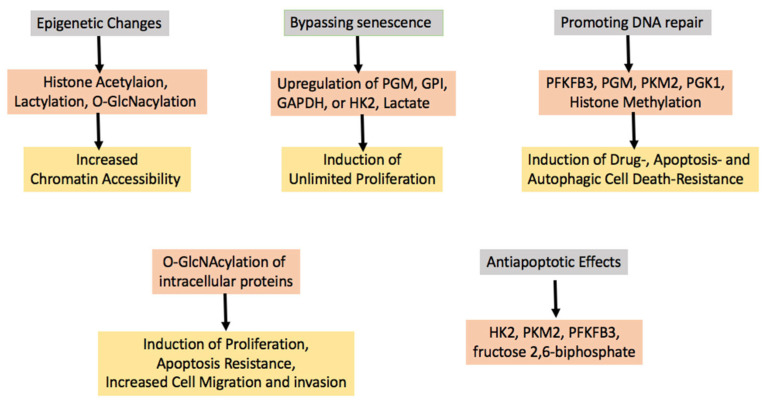
Mechanisms, Mediators, and Final Effects of Glycolysis-Induced Tumorigenesis. The figure depicts the mechanisms involved in glycolysis-induced tumorigenesis (grey-shaded boxes), the glycolytic enzymes or metabolites that act as mediators (pink-shaded boxes), and the final effects that are induced in cells undergoing the tumorigenic process (yellow-shaded boxes). Abbreviations: GAPDH, glyceraldehyde 3-phosphate dehydrogenase; GPI, glucose 6-phosphate isomerase; HK, hexokinase; O-GlcNac, *O*-linked β-*N*-acetylglucosamine; PFKFB3, 6-phosphofructo-2-kinase/fructose-2,6-biphosphatase 3; PGM, phosphoglycerate mutase; PGK, phosphoglycerate kinase; PKM2, pyruvate kinase M2.

**Figure 3 cells-12-01124-f003:**
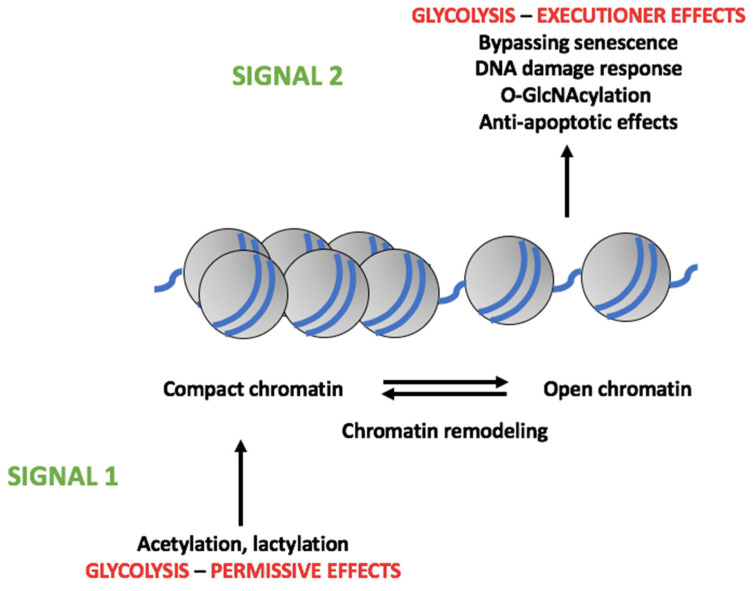
Effects on Early Tumorigenesis Induced by Glycolytic Enzymes and Metabolites. Glycolytic metabolites (e.g., pyruvate and lactate) may induce chromatin remodeling as a result of histone acetylation and lactylation. We refer to these as permissive effects or signal 1. These effects are necessary for the induction of what we call executioner effects or signal 2, i.e., the eventual mediators of the biological effects instigating tumorigenesis: bypassing senescence, DNA damage response, O-GlcNAcylation, anti-apoptotic effects.

**Figure 4 cells-12-01124-f004:**
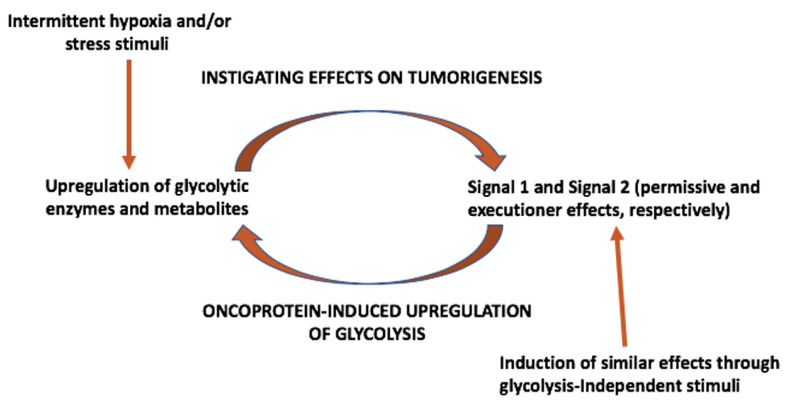
Feed-Forward Amplification Loop between Upregulated Glycolysis and Oncoproteins in Promoting Early Tumorigenesis. Intermittent hypoxia and/or other stress stimuli can induce the upregulation of glycolysis and such upregulation may induce Signal 1 and/or Signal 2 (permissive and executioner effects, respectively) as illustrated in Figure 3. Oncoproteins may, by themselves, upregulate glycolysis, thereby making glycolytic enzymes and metabolites available for the renewed induction of oncogenic signaling mechanisms. This feed-forward regulation may be at the basis of the multistep process that is known to be required to achieve full-blown tumor growth. We suggest that permissive and executioner effects can be induced by other glycolysis-independent stimuli.

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
