# Peer review of "On the Role of Glycolysis in Early Tumorigenesis—Permissive and Executioner Effects"

_cells, 2023, doi:10.3390/cells12081124_

Round 1

Reviewer 1 Report

The manuscript by Marcucci F and Rumio C with the title “On the Role of Upregulated Glycolysis in Early Tumorigenesis” is interesting due to the importance of glycolysis as an energy fuel for tumor growth. However there are some questions and comments related to the content of the manuscript.

-In my opinion, the title could be “The Role of Glycolysis in Tumorigenesis” because it is not clear if all cancers analyzed in the manuscript were reported in the early stages of cancer development.

-The authors should check the phrases because there are some grammatical errors and typos in the manuscript.

Abstract section

The phrase “In this article we summarize the evidence suggesting that upregulated glycolysis may be involved in tumor initiation and,…….” It could be restructured as “ ….. In this article we summarize the evidence that upregulated glycolysis is involved in tumor initiation and …”

Introduction section

-Please check the references to confirm that they are in only format with all data necessary.

-In the introduction section, the authors should add information about the following subtitles: chromatin remodeling, senescence, proliferation, DNA repair, O-linked N-acetylglucosamine modification, antiapoptotic effects, epithelial-mesenchymal transition, autophagy, and angiogenesis focused with glycolysis.

-Figure 1, in this figure it is necessary to add the following: glucose-6-phosphate to ribulose-5- phosphate generates 2 NADPH. Please check in figure 1, which reactions are irreversible and reversible in the glycolysis pathway, and please add the cofactors for each reaction when necessary. In figure legend, it should be mentioned that the pentose phosphate pathway is summarized. Please in each reaction, add metabolites incorporated and metabolites produced (e.g., ATP, Acetil-coA, CO2, etc). When piruvato is transformed to Acetil-coA (add aerobic glycolysis) or pyruvate to lactate (add anaerobic glycolysis).

-The following phrase: “Moreover, upregulated glycolysis can also increase the fueling of metabolites to anabolic metabolism, including the pentose phosphate pathway (PPP), which allow cells to synthesize precursors for biological macromolecules (e.g., nucleotides, amino acids, and lipids) 31, 32]”. It should be related with figure 1.

2. Clinical Evidence for a Role of Glycolysis in Early Tumorigenesis. In this section, the authors should considered only the type’s cancers reported the glycolysis in early tumorigenesis.

-In the phrase (page 4, line 110-112): First of all, clinical conditions characterized by hyperglycemia like type II diabetes and obesity are associated with an increased incidence of cancer [33, 34]. Please add examples of cancers associated with type II diabetes and obesity.

-Please in all the manuscript, the reference must be indicated at the end of phrases because is confused with the next reference which belongs to another phrase. For instance, “Of note, strong GLUT1 expression was highly specific for detecting gallbladder carcinomas.” Here is necessary to add the reference 39.

-In the following phrase (page 5, line 165-166): “KLF4 had been previously identified as an essential factor for pluripotent stem cell development [45].” The reference 45 is not appropriate for the information considered.

-In the following phrase (page 5, line 165-166): “ and induction of aerobic glycolysis [45].” Please check if the interpretation is correct, because in the reference 45 it says “KLF4 overexpression significantly attenuated the aerobic glycolysis in and growth of pancreatic cancer cells…..”

3. Oncoproteins and Other Oncogenic Molecules Are Potent Upregulators of Glycolysis

-In page 5 line 186: “epidermal growth factor receptor (EGFR) [57, 58]” Please check the reference 57 if it is correct according to information considered.

-In page 5 line 187:and human epidermal growth factor receptor (HER) 2 [31].” The reference 31 is not appropriate for the information considered.

-In page 7 line 273: “and inhibit glycolysis through various mechanisms [79-81].” Please add as example some mechanisms involved.

- In page 7 line 274: “p53 is a positive regulator of mitochondrial metabolism and is involved in cellular redox control [81].” The authors should check if the reference 81 is correct.

-I recommend another figure considering the most important metabolites generated by glycolysis and molecular pathways that have effect on chromatin remodeling, senescence, proliferation, DNA repair, and so on.

5. Mechanistic Aspects of the Involvement of Glycolysis in Early Tumorigenesis

-In page 11, line 464-465:Glycolytic metabolism was found to increase chromatin accessibility by inducing global histone acetylation [115],” Please recheck reference 115 in order to confirm that it is correct.

-In page 13, line 554-555:Eventually, DDR-promoting effects of both upregulated glycolysis and glutaminolysis have been demonstrated in irradiated MCF7 breast cancer cells [131].” The authors should check the reference 131 in order to confirm that the information considered in the manuscript is correct.

Conclusion

In the phrase. “In this article we have summarized available evidence on the role glycolysis in early tumorigenesis.” Please consider if all cancers analyzed were in in early tumorigenesis.

Author Response

Beforehand we would like to thank the referee for his/her meticulous reviewing and for havins spent so much time for analyzing the manuscript.

Criticism: In my opinion, the title could be “The Role of Glycolysis in Tumorigenesis” because it is not clear if all cancers analyzed in the manuscript were reported in the early stages of cancer development. 

Answer: This was a difficult criticism to address because the two referees had divergent views on this point. Referee 1 proposed a more concise title, while referee 2 proposed a more descriptive title. We propose to change the title as follows: On the Role of Glycolysis in Tumorigenesis – Permissive and Executioner Effects. In fact, the term “Tumorigenesis”, which encompasses the terms Tumor and Genesis relates, in fact, to the steps that lead, progressively, to the generation of a tumor rather than to its growth. Apart from this, we confirm that most, if not all, articles that have been cited in the manuscript related to the effects of glycolysis on neoplastic transformation and tumorigenesis.

Criticism: The authors should check the phrases because there are some grammatical errors and typos in the manuscript.

Answer: We have gone through the whole text and amended several errors.

Criticism: The phrase “In this article we summarize the evidence suggesting that upregulated glycolysis may be involved in tumor initiation and,…….” It could be restructured as “ ….. In this article we summarize the evidence that upregulated glycolysis is involved in tumor initiation and …”

Answer: We have changed the sentence as suggested by the referee.

Criticism: Please check the references to confirm that they are in only format with all data necessary. 

Answer: We have checked all references.

Criticism: In the introduction section, the authors should add information about the following subtitles: chromatin remodeling, senescence, proliferation, DNA repair, O-linked N-acetylglucosamine modification, antiapoptotic effects, epithelial-mesenchymal transition, autophagy, and angiogenesis focused with glycolysis.

Answer: Although this request was not entirely clear to us, we have now added in the Introduction a new paragraph with a very concise explanation of each of the processes that have been mentioned by the referee. We hope that this addresses satisfactorily the criticism. 

Criticism: Figure 1, in this figure it is necessary to add the following: glucose-6-phosphate to ribulose-5-phosphate generates 2 NADPH. Please check in figure 1, which reactions are irreversible and reversible in the glycolysis pathway, and please add the cofactors for each reaction when necessary. In figure legend, it should be mentioned that the pentose phosphate pathway is summarized. Please in each reaction, add metabolites incorporated and metabolites produced (e.g., ATP, Acetil-coA, CO2, etc). When piruvato is transformed to Acetil-coA (add aerobic glycolysis) or pyruvate to lactate (add anaerobic glycolysis). 

Answer: The referee is right as regards the fact that 2 molecules of NADPH are produced during the transformation of G6P to ribulose 5P. Yet, we have decided to delete NADPH overall since we realized that if we left it, we should also add the NADH molecules that are produced during glycolysis. Please consider that this figure is not intended to offer an exact view of the biochemical pathways that are depicted but, rather, a simplified view able to support the reader in the comprehension of the text. If, however, a detailed description of the whole glycolytic pathway is deemed necessary we can certainly add it, although we fear that such a complex figure might be a bit confusing for the reader. In any case, we have now indicated which reactions are irreversible (green arrows) and which steps occur under aerobic conditions (aerobic glycolysis) or under anaerobic condition and some special conditions occurring in the presence of aerobiosis (anaerobic glycolysis).

Criticism: The following phrase: “Moreover, upregulated glycolysis can also increase the fueling of metabolites to anabolic metabolism, including the pentose phosphate pathway (PPP), which allow cells to synthesize precursors for biological macromolecules (e.g., nucleotides, amino acids, and lipids) 31, 32]”. It should be related with figure 1.

Answer: This sentence is now related to Fig. 1, as suggested by the reviewer.

Criticism: In the phrase (page 4, line 110-112): First of all, clinical conditions characterized by hyperglycemia like type II diabetes and obesity are associated with an increased incidence of cancer [33, 34]. Please add examples of cancers associated with type II diabetes and obesity.

Answer: The tumor types have now been added, as requested by the referee.

Criticism: Please in all the manuscript, the reference must be indicated at the end of phrases because is confused with the next reference which belongs to another phrase. For instance, “Of note, strong GLUT1 expression was highly specific for detecting gallbladder carcinomas.” Here is necessary to add the reference 39. 

Answer: We have now gone through the whole manuscript and, whenever possible, we have now shifted the references to the end of the sentences where they are discussed. We added Ref. 39 at the end of the sentence cited by the referee.

Criticism: In the following phrase (page 5, line 165-166): “KLF4 had been previously identified as an essential factor for pluripotent stem cell development [45].” The reference 45 is not appropriate for the information considered.

Answer: We have now replaced the former reference 45 with a new one.

Criticism: In the following phrase (page 5, line 165-166): “ and induction of aerobic glycolysis [45].” Please check if the interpretation is correct, because in the reference 45 it says “KLF4 overexpression significantly attenuated the aerobic glycolysis in and growth of pancreatic cancer cells…..”

Answer: see previous answer.

Criticism: In page 5 line 186: “epidermal growth factor receptor (EGFR) [57, 58]” Please check the reference 57 if it is correct according to information considered. 

Answer: Yes, it is correct. We have checked the article.

Criticism: In page 5 line 187: “and human epidermal growth factor receptor (HER) 2 [31].” The reference 31 is not appropriate for the information considered.

Answer: The reference is appropriate. In fact, in the abstract of the article it is stated: “we showed that HK2 is required for tumor initiation and maintenance in mouse models of KRas-driven lung cancer, and ErbB2-driven breast cancer, despite continued HK1 expression.” In general, I have tried to select as references those articles which address also glycolysis. This is in order to keep the reference list within reasonable limits. On the other hand, we had now added to the next, next to HER2 also the ErbB2 abbreviation.

Criticism: In page 7 line 273: “and inhibit glycolysis through various mechanisms [79-81].” Please add as example some mechanisms involved.

Answer: We have now extended the sentence by adding some of the mechanisms on the negative control of p53 on glycolysis.

Criticism:  In page 7 line 274: “p53 is a positive regulator of mitochondrial metabolism and is involved in cellular redox control [81].” The authors should check if the reference 81 is correct. 

Answer: After having checked, we agree that this is not an appropriate reference. We have added a new which has now become ref. 82.

Criticism: I recommend another figure considering the most important metabolites generated by glycolysis and molecular pathways that have effect on chromatin remodeling, senescence, proliferation, DNA repair, and so on. 

Answer: We have now added a new figure, Figure 2 of the revised version.

Criticism: In page 11, line 464-465: “Glycolytic metabolism was found to increase chromatin accessibility by inducing global histone acetylation [115],” Please recheck reference 115 in order to confirm that it is correct. 

Answer: Yes, it is correct. We have checked the article.

Criticism: In page 13, line 554-555: “Eventually, DDR-promoting effects of both upregulated glycolysis and glutaminolysis have been demonstrated in irradiated MCF7 breast cancer cells [131].” The authors should check the reference 131 in order to confirm that the information considered in the manuscript is correct. 

Answer: Yes, it is correct. We have checked the article.

Criticism. Conclusion. In the phrase. “In this article we have summarized available evidence on the role glycolysis in early tumorigenesis.” Please consider if all cancers analyzed were in in early tumorigenesis.

Answer: Please see our reply to the first criticism to this manuscript.

Reviewer 2 Report

The present manuscript presents well documented evidence on the role of upragulated glycolysis in early tumorgenesis, describing the clinical evidence, the role of oncoproteins and oncosuppresive proteins, the causative role of glycolysis in tumorigenesis. The article also proposes an integrative model for explaining how permissive and executioner effects of glycolysis can initiate the tumor process.

The manuscript is generally well written and it is well organized. However, I have several observations and recommendation which aim to improve even more the current form:

Lines:

16, 69-70, 159-160, 318-319, 473-474, 481-482, 488-489, 578-579, 630, 690-691 – please rephrase/correct to improve the readability of the text.

78- it is unusual for antitumor therapeutics to cause the switch to a glycolytic phenotype. Perhaps it is related to drug resistance? Please explain / clarify.

92 – Please include a verb - “Beforehand, however, some preliminary considerations”

164-165 – Please rephrase to avoid plagiarism

196 - Please revise weather it is not the KRAS oncoprotein which interacts with HK1, as you describe it as a post-translational modification

243-247 - Perhaps this paragraph can be moved after line 235, as it discusses the impact of PI3K-Akt-mTOR pathway on the same glycolitic enzyme as before, PKM2.

281-285, 642-646 – Perhaps you could consider splitting the phrase for a clearer understanding of the text.

434 – There are two end points.

477 – Has instead of ahs.

524-526 - Please rephrase to improve the readability/fluency of the text or consider splitting.

572 – There are two “in”.

577 – Please include full name for RAP1.

136, 158, 193, 283, 310, 393, 522, 577, 755 – missing commas, please add them into the right places.

Figure 1 – The resolution can be increased and more attention can be paid to the alignment of the arrows.

Figure 2 – From the direction of the present arrow it looks like signal 2 effects (the executioner ones) can impact chromatin remodeling. But if I well understood your proposed model, signal 2 effects can be triggered by / be a consequence of signal 1 effects. If so, shouldn't be also an arrow pointing up, from remodeled chromatin to signal 2 effects?

Figure 3 – Perhaps figure 3, along with the discussion related, can be included in chapter 6 “A Mechanistic Model for Glycolysis-Induced Tumorigenesis”, at the end, or as a distinct subchapter? Usually in the conclusion part you should have clear, take home messages, to briefly summarize the content.

As a suggestion - perhaps the title of the article should be slightly more descriptive, suggesting more precisely what is being discussed in the manuscript

In conclusion, I congratulate the authors for their manuscript and, after revising for the suggested changes, the manuscript can be accepted for publication.

Author Response

Beforehand we would like to thank the referee for his/her nice comments to our manuscript, for the meticulous reviewing and for having spent so much time with our manuscript.

Criticism: Lines: 16, 69-70, 159-160, 318-319, 473-474, 481-482, 488-489, 578-579, 630, 690-691 – please rephrase/correct to improve the readability of the text.

Answer: We have changed the lines as suggested by the reviewer.

Criticism: 78- it is unusual for antitumor therapeutics to cause the switch to a glycolytic phenotype. Perhaps it is related to drug resistance? Please explain / clarify.

Answer: We don’t know if we have understood this criticism right. We have cited one article showing that a chemotherapeutic drug upregulates. In a previous review of ours (Neoplasia 2021;23:234-245) we have cited quite a number of articles showing that antitumor drugs of several classes can upregulate glycolysis (see references 18 , 49 , 73 , 91, 47, 82 , 92, 59 in the Neoplasia paper). While this is certainly not a universal phenomenon, we believe that there is sufficient evidence in support of this statement. For this reason we propose to leave this line as it currently stands

Criticism: 92 – Please include a verb - “Beforehand, however, some preliminary considerations”

Answer: Done.

Criticism: 164-165 – Please rephrase to avoid plagiarism.

Answer: Done

Criticism: 196 - Please revise weather it is not the KRAS oncoprotein which interacts with HK1, as you describe it as a post-translational modification.

Answer: We have rephrased the sentence as follows: This is when glycolytic enzymes undergo a posttranslational modification or an interaction with other molecules that endows them with enhanced activity.

Criticism: 243-247 - Perhaps this paragraph can be moved after line 235, as it discusses the impact of PI3K-Akt-mTOR pathway on the same glycolitic enzyme as before, PKM2. 

Answer: Done.

Criticism: 281-285, 642-646 – Perhaps you could consider splitting the phrase for a clearer understanding of the text.

Answer: Done

Criticism: 434 – There are two end points.

Answer: Rectified.

Criticism: 477 – Has instead of ahs.

Answer: Rectified.

Criticism: 524-526 - Please rephrase to improve the readability/fluency of the text or consider splitting.

Answer: Done.

Criticism: 572 – There are two “in”.

Answer: Rectified.

Criticism: 577 – Please include full name for RAP1.

Answer: Done.

Criticism: 136, 158, 193, 283, 310, 393, 522, 577, 755 – missing commas, please add them into the right places.

Answer: Done.

Criticism: Figure 1 – The resolution can be increased and more attention can be paid to the alignment of the arrows.

Answer: We have improved the alignment of the arrows. We have changed the figure with one tha should have a better resolution.

Criticism: Figure 3 – Perhaps figure 3, along with the discussion related, can be included in chapter 6 “A Mechanistic Model for Glycolysis-Induced Tumorigenesis”, at the end, or as a distinct subchapter? Usually in the conclusion part you should have clear, take home messages, to briefly summarize the content.

Answer: We have followed the suggestion of the referee. Figure 4 (former figure 3) and the corresponding text are now under chapter 6.

Criticism: As a suggestion - perhaps the title of the article should be slightly more descriptive, suggesting more precisely what is being discussed in the manuscript.

Answer: Answer: This was a difficult criticism to address because the two referees had divergent views on this point. Referee 1 proposed a more concise title, while referee 2 proposed a more descriptive title. We propose to change the title as follows: On the Role of Glycolysis in Tumorigenesis – Permissive and Executioner Effects. In fact, the term “Tumorigenesis”, which encompasses the terms Tumor and Genesis relates, in fact, to the steps that lead, progressively, to the generation of a tumor rather than to its growth.

Round 2

Reviewer 1 Report

I have no comments for the authors. I gree to the changes. 

Reviewer 2 Report

I agree the publication of the manuscript in the revised form.